# The Effect of Sodic Water Type on the Chemical Properties of Calcareous Soil in Semi-Arid Irrigated Land

Ayşe E. Peker [1], Hasan S. Öztürk [2],* and Amrakh I. Mamedov [3]

[1] Black Sea Agricultural Research Institute, Samsun 55300, Turkey; ayse.ertaspeker@tarimorman.gov.tr
[2] Department of Soil Science and Plant Nutrition, Faculty of Agriculture, Ankara University, Ankara 06110, Turkey
[3] Faculty of Agriculture, Arid Land Research Center, Tottori University, Tottori 680-0001, Japan; amrakh03@yahoo.com
* Correspondence: hozturk@agri.ankara.edu.tr

**Abstract:** Irrigation of calcareous soil with saline–sodic water can modify the composition of the soil solution and exchange complexes in agricultural land of arid and semi-arid regions with low water resources. The objective of this study was to monitor (medium-term) potential changes in a calcareous clay soil irrigated with two types of sodic waters without cropping. Irrigation water with two high sodium adsorption ratios (SAR = 20 and 40) and electrical conductivity (EC < 3 dS m$^{-1}$) was prepared using NaCl and NaHCO$_3$ salts. The sodic irrigation waters were applied (June–October) in three periods (1, 2, and 4; one period = five irrigations) to bare non-saline soil with drip irrigation during two growing seasons; no irrigation action was taken in the winter–spring rainy season (period 3). Sampling (0–30 cm) was made after each period to determine the changes in soil pH, EC, water-soluble Na$^+$, Ca$^{2+}$, Mg$^{2+}$, K$^+$, Cl$^-$, and HCO$_3{}^-$. Relative to the control, irrigation with both sodic waters increased soil pH, EC, and water-soluble Na$^+$ and decreased or did not change water-soluble cations (Ca$^{2+}$, Mg$^{2+}$). The Cl$^-$ concentration increased rapidly with NaCl-type water application, but it was leached away quickly by winter–spring rains. The HCO$_3{}^-$ concentration increased with NaHCO$_3$-type water application, yet it leached out slowly in the rainy period. The movement of HCO$_3{}^-$ ions in the upper soil profile (0–30 cm) was significantly slower compared to Cl$^-$ ions. Dissolution of slightly soluble soil CaCO$_3$ by irrigation increased the solution concentration of Ca$^{2+}$ and its mobility, yet the kinetics of processes depended on water type and irrigation period. The released Ca$^{2+}$ interacted with other cations in the soil, causing further significant positive physicochemical changes in the soil solution and exchange capacity (comparable with control soil) at the end of the irrigation period. The CaCO$_3$ content in the soil would be a long-term guarantee of the Ca$^{2+}$ resource in soils, even if the amount of water-soluble Ca$^{2+}$ may decrease for the short-term period during irrigation. The results should be considered for rational irrigation management (with various water qualities) in semi-arid and arid regions.

**Keywords:** land use; sodic water; ion exchange; ion transport; semi-arid and arid region

## 1. Introduction

Almost 40% of global agricultural production comes from just 20% of the world's irrigated farmland, showing that irrigation, along with other management, has more than doubled land productivity in arid conditions [1]. However, due to the priority given to water resources in urban areas in arid and semi-arid regions, freshwater resources suitable for irrigation rapidly decrease in these climatic regions [2]. Moreover, one-third of the world's agricultural lands are affected by soil degradation associated with erosion and salinity [3]. Therefore, applying irrigation with lower water quality (e.g., treated wastewater, saline water, effluent, drainage, and groundwater) to agricultural land is common in such regions. However, irrigation water quality and efficiency must be considered to prevent secondary

salinization in the soil and a more significant reduction in soil quality [4,5]. It is undisputed that salinity and alkalinity are the most accepted water quality parameters of concern. Salinity refers to the total concentration of dissolved salts in soil and water, whereas water quality largely depends on the composition and concentration of dissolved ions or salts [6–8]. Using saline irrigation water causes salt accumulation in the soil, and salts containing $Na^+$ ions (e.g., NaCl, $NaHCO_3$) cause salt or abiotic stress (e.g., decreased transpiration, specific toxic ion effect) to the plants and deteriorate soil structure and physical properties and quality [9,10]. However, soil degradation caused by salinization can be suppressed by the calcium carbonate ($CaCO_3$) in the soil, especially in arid regions. Therefore, depending on water quality, management, and climatic conditions, its deteriorative effect may appear in more extended periods. This, unfortunately, enables local farmers to continue using these waters without proper consideration of affecting factors [7–10].

In arid and semi-arid regions, salts are mainly found in irrigation water in the form of chlorides ($Cl^-$), sulfates ($SO_4^{2-}$), and bicarbonates ($HCO_3^-$) of calcium ($Ca^{2+}$), magnesium ($Mg^{2+}$), sodium ($Na^+$), and potassium ($K^+$) [11]. Saline–sodic irrigation waters can significantly increase soil sodicity in dry regions with limited rainfall and high evaporation; for example, drip irrigation with saline–sodic water (EC = 3.0–8.5 dS $m^{-1}$ and SAR = 14–26 $mmol^{1/2}$ $L^{-1/2}$) significantly increased soil sodicity during the short period under cotton production [12]. The electrical conductivity (EC) of the soil solution, water-soluble Na, and Cl ion content in the soil increased with the increase in the salinity of the irrigation water in two years [13]. In other studies, excessive exchangeable Na with pH > 8.5 in irrigation water caused the physical properties of soils to deteriorate and adversely affected water and air movement, soil erodibility, and plant growth [14–16]. Irrigation with alkaline waters (with high carbonate and bicarbonate content) caused an increase in soil pH and Na saturation of the soils. As a result, notable soil aeration and permeability reduction were noted due to soil aggregate slaking and clay dispersion and clogging soil pores [17,18]. Yet, the result was soil texture dependent [19,20]. In similar experimental conditions, crop productivity was negatively affected by soil salinity and sodicity [21–23].

In arid and semi-arid regions, the ion compositions are changed in the exchange complexes and soil solution after being irrigated with sodic water. Sodic water usage in non-saline soil for short periods (e.g., up to 2–3 irrigation periods) is often necessary due to water scarcity. High clay and $CaCO_3$ content in a calcareous clay soil limit the movement of $Na^+$ ions downwards and structure deterioration [24,25]. With the dissolution of $CaCO_3$ during the use of saline water, there is an increase in exchangeable $Ca^{2+}$ in the calcareous clay soil, which competes with the exchangeable $Na^+$ in the soil solution, thus limiting the increase in soil exchangeable $Na^+$ and SAR in the soil solution or solid phase [26]. Furthermore, elevated soil salinity and sodicity can affect soil hydraulics by changing the pore size distribution through clay dispersion and flocculation [9,27]. Results of numerical studies from Turkiye and foreign countries were in the same line, reporting that, along with irrigation water quality, factors such as soil and salt type, irrigation frequency and season, and soil permeability (linked to soil texture and type) should be considered in evaluating the potential of irrigation water quality on physicochemical properties and quality of soils in arid and semi-arid lands [28–32].

The overall water demand in Turkiye continues to increase, and it is expected that the country will suffer from water scarcity in the following decades, particularly in arid and semi-arid territories, as a result of population growth and the impact of climate change [33–35]. Approximately 74% of the total water supply is used for agricultural irrigation, and 11% and 15% are used for industrial and domestic purposes, respectively. Therefore, the short- and long-term contributions of water quality to soil properties and land quality must be considered as a high-priority problem to tackle in the regions with scarce water resources. The quality of water in the area where the study was conducted is mostly not suitable for irrigation, as it may contain significant amounts of sodium. The objective of this study was to evaluate the effect of sodic water type (quality) on the chemical properties of a calcareous clay soil in semi-arid irrigated land to monitor the potential of medium-term changes in

the ion dynamics. A typical calcareous soil was chosen to prove to what extent calcium carbonate can tolerate the adverse effects of sodic water.

## 2. Material and Methods

### 2.1. Location

This study was conducted as a field trial at Ankara University Research and Application Farm in Haymana, Ankara (Figure 1), in a semi-arid region of Central Anatolia over two years. The distinctive feature of the region's soils is low organic matter content but high amounts of carbonate, mostly calcium carbonate. For latter reason, aggregate formation and relatively strong soil structure are evident. The study area is located at an average altitude of 1065 m above sea level and has a continental climate zone characterized by little precipitation and significant temperature variations. In the Haymana district, summers are warm, dry, and open, and winters are freezing, snowy, and partly cloudy. The temperature typically ranges from $-6$ °C to 29 °C throughout the year, rarely below $-13$ °C or above 33 °C. The average temperature in the province is 11.7 °C, and the annual average precipitation is 390 mm. The evaporation demand is high, especially between April and October (up to 1222 mm) [25].

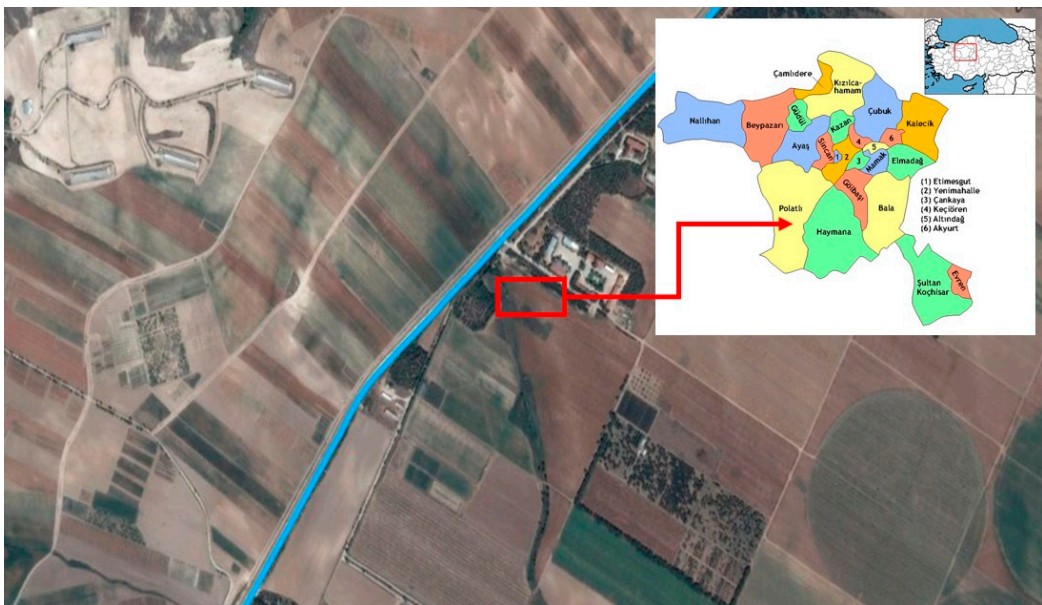

**Figure 1.** Location of the experimental plots.

### 2.2. Experimental Design

The experimental design of field plots (3 × 4 m and 2 m apart from each other) was randomized blocks with three replications: drip irrigation method with (i) control (fresh water, EC < 1 dS m$^{-1}$) was used, and (ii) four sodic waters (NaCl SAR 20, NaCl SAR 40, NaHCO$_3$ SAR 20 and NaHCO$_3$ SAR 40) were applied (5 treatments × 3 replication = 15 plots). Sodic waters (EC < 3 dS m$^{-1}$) with two SAR levels (20 and 40) were prepared using NaCl and NaHCO$_3$ salts; the ion concentrations of the irrigation water were computed with the Extract Chem Software program [36]. The plots were not cultivated and not cropped. Pesticides were used for weed control. Bare plots were irrigated for two years during the June–October period. The initial soil characteristics (pH, EC, water-soluble Na$^+$, Ca$^{2+}$, Cl$^-$, and HCO$_3$$^-$ contents) were used as a control to evaluate the contribution of irrigation with sodic water quality on soil properties (Figures 2–7). Groundwater depth was >20 m, and thus, did not affect the dynamics of soil salinity.

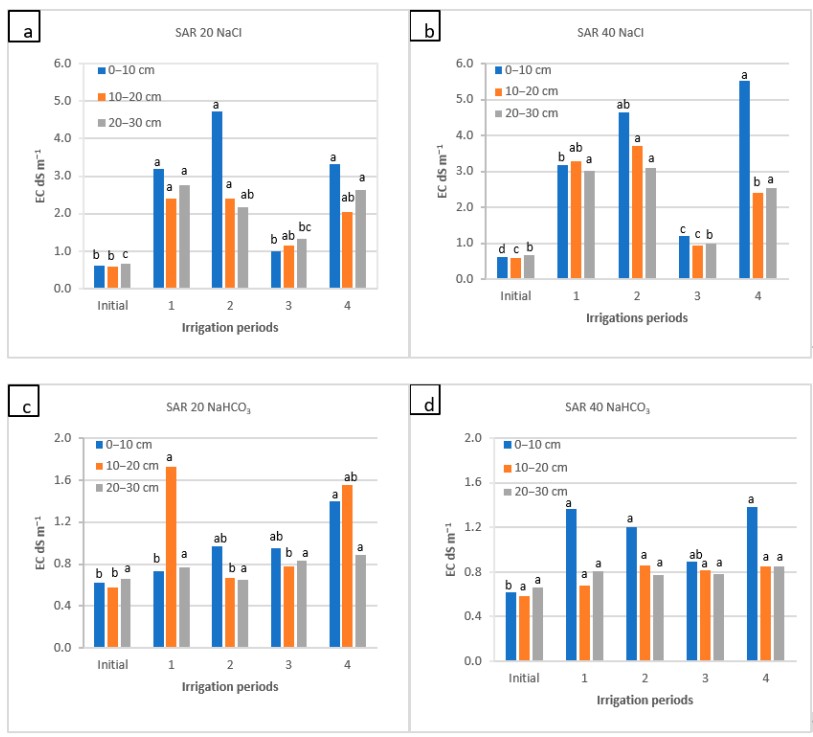

**Figure 2.** Soil $EC_e$ as a function of water application during the irrigation periods (1, 2, and 4) and the rainy period (3) for four water qualities: (**a**) NaCl water: SAR 20; (**b**) NaCl water: SAR 40; (**c**) NaHCO$_3$ water: SAR 20; and (**d**) NaHCO$_3$ water: SAR 40. For each water quality, within each soil depth, the columns labeled with the same letter are not significantly different at $p < 0.05$.

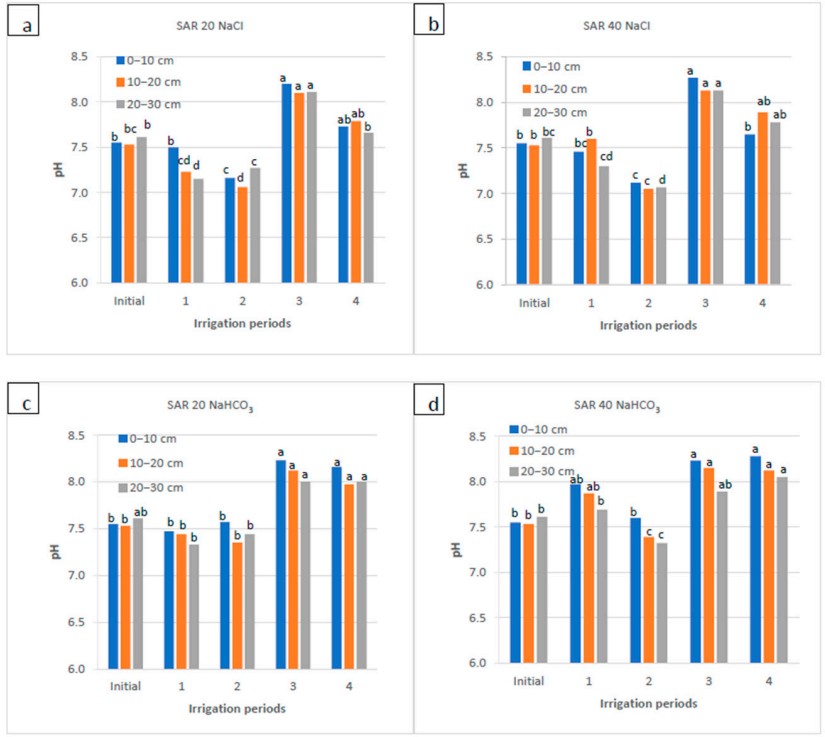

**Figure 3.** Soil pH as a function of irrigation water applications during the irrigation periods (1, 2, and 4) and the rainy period (3) for four water qualities: (**a**) NaCl water: SAR 20; (**b**) NaCl water: SAR 40; (**c**) NaHCO$_3$ water: SAR 20; and (**d**) NaHCO$_3$ water: SAR 40. For each water quality, within each soil depth, the columns labeled with the same letter are not significantly different at $p < 0.05$.

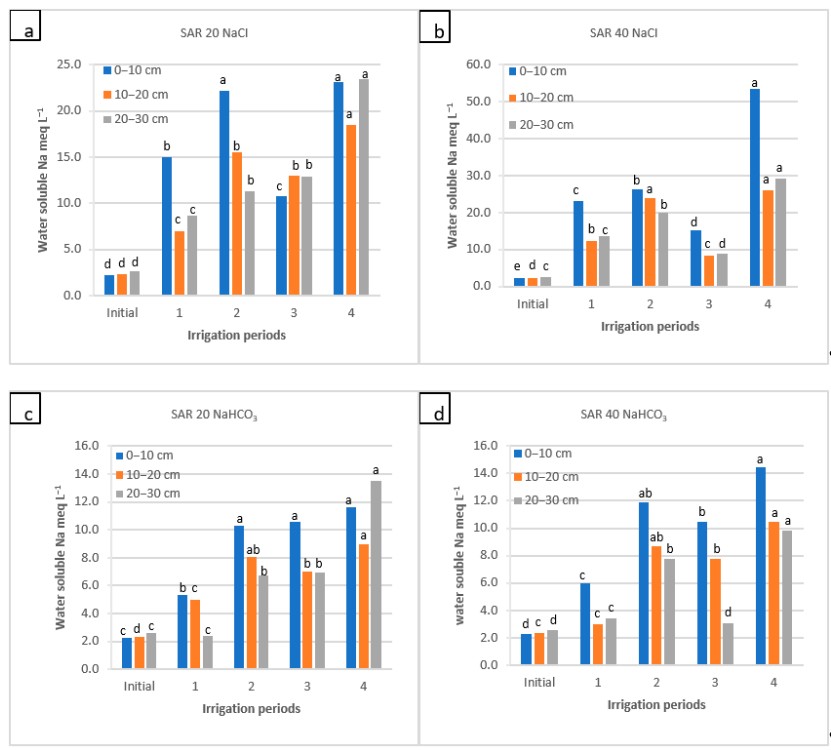

**Figure 4.** Water-soluble Na$^+$ of soil as a function of irrigation water applications during the irrigation periods (1, 2, and 4) and the rainy period (3) for four water qualities: (**a**) NaCl water: SAR 20; (**b**) NaCl water: SAR 40; (**c**) NaHCO$_3$ water: SAR 20; and (**d**) NaHCO$_3$ water: SAR 40. For each water quality, within each soil depth, the columns labeled with the same letter are not significantly different at $p < 0.05$.

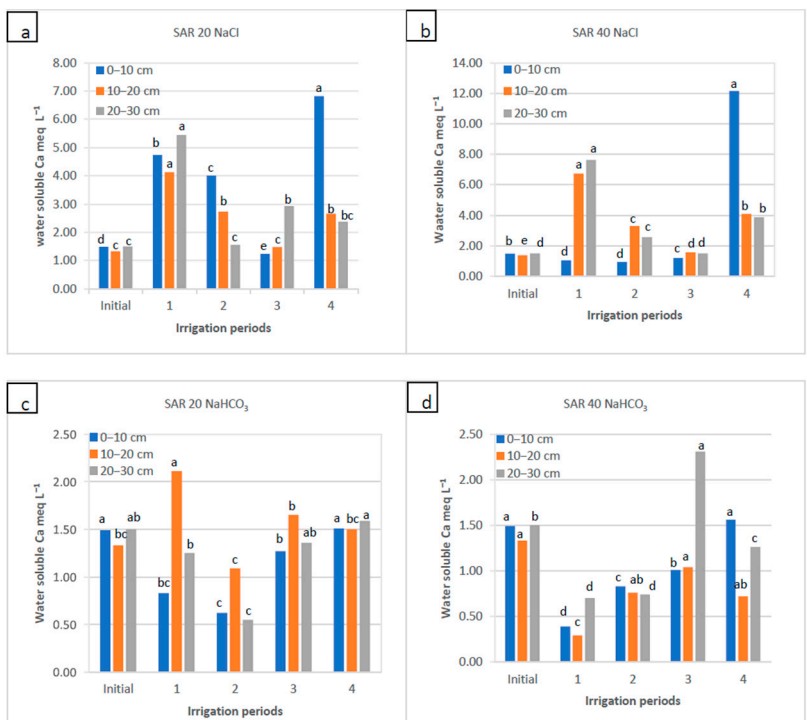

**Figure 5.** Water-soluble Ca$^{2+}$ of soil as a function of irrigation water applications during the irrigation periods (1, 2, and 4) and the rainy period (3) for four water qualities: (**a**) NaCl water: SAR 20; (**b**) NaCl water: SAR 40; (**c**) NaHCO$_3$ water: SAR 20; and (**d**) NaHCO$_3$ water: SAR 40. For each water quality, within each soil depth, the columns labeled with the same letter are not significantly different at $p < 0.05$.

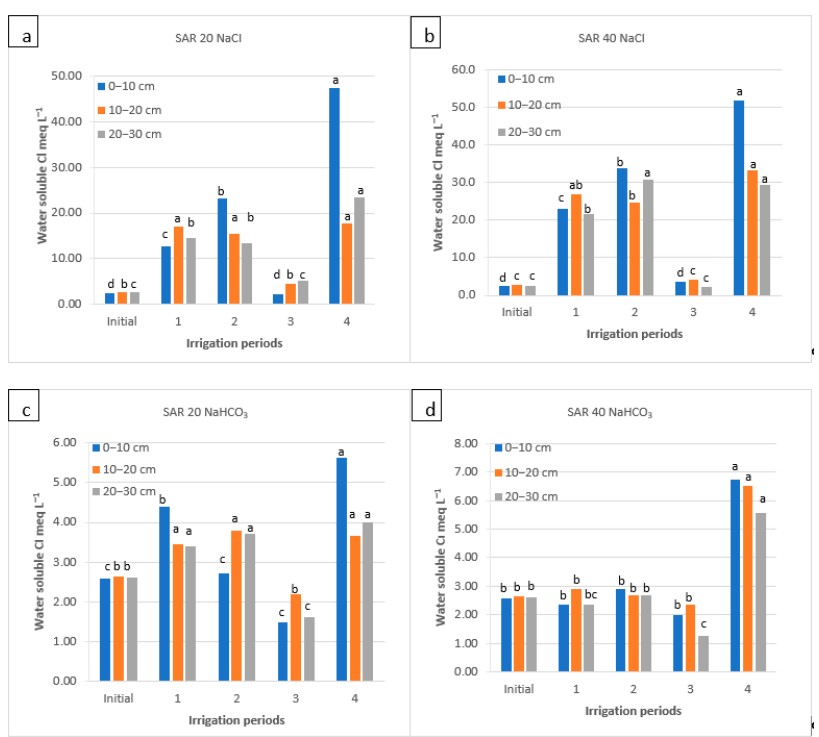

**Figure 6.** Water-soluble $Cl^-$ of soil as a function of irrigation water applications during the irrigation periods (1, 2, and 4) and the rainy period (3) for four water qualities: (**a**) NaCl water: SAR 20; (**b**) NaCl water: SAR 40; (**c**) NaHCO$_3$ water: SAR 20; and (**d**) NaHCO$_3$ water: SAR 40. For each water quality, within each soil depth, the columns labeled with the same letter are not significantly different at $p < 0.05$.

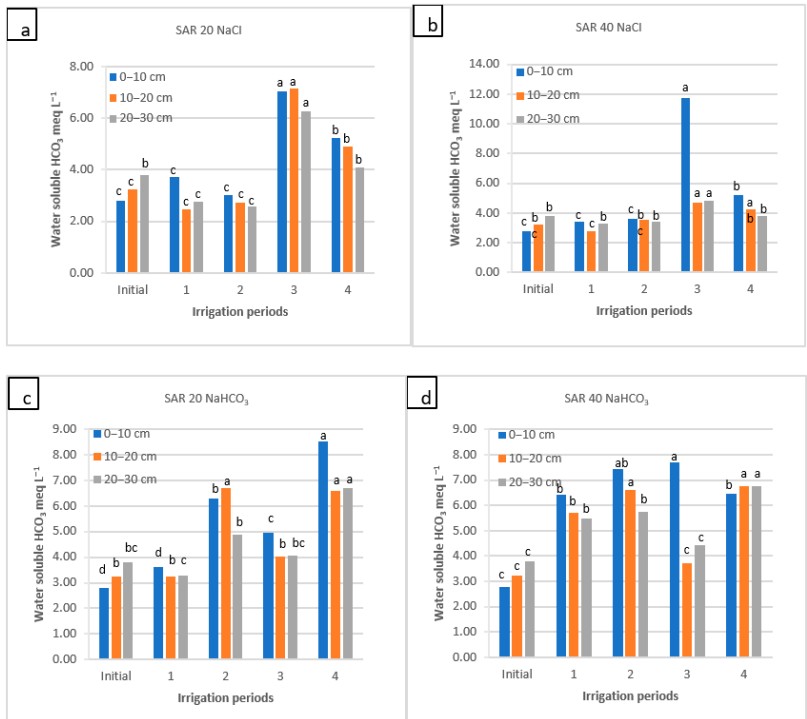

**Figure 7.** Water-soluble $HCO_3^-$ of soil as a function of irrigation water applications during the irrigation periods (1, 2, and 4) and the rainy period (3) for four water qualities: (**a**) NaCl water: SAR 20; (**b**) NaCl water: SAR 40; (**c**) NaHCO$_3$ water: SAR 20; and (**d**) NaHCO$_3$ water: SAR 40. For each water quality, within each soil depth, the columns labeled with the same letter are not significantly different at $p < 0.05$.

Details of irrigation with sodic water and soil sampling are provided in Table 1. The irrigations were carried out during four periods: periods 1, 2, and 4 were irrigation periods (one period = five irrigations), and period 3 was the winter–spring (October–May) precipitation period (Table 1). This is the period when the soil leaching occurred. The total precipitations that fell were 338.0, 32.2, 276.2 mm, and 190.8 mm during the 1st, 2nd, 3rd, and 4th periods, respectively. For each sodic water type's treatment, the irrigation rate applied to the plots was based on soil moisture content and was ~10% greater than soil field capacity to avoid obstacles with potential inaccuracy and deep infiltration (based on the preliminary tests). A soil moisture sensor was used to control the soil moisture content (ECH 20 EC-5; METER Group, Inc., Pullman, WA, USA). Irrigation was repeated when the soil moisture decreased to the wilting point. During the 3rd period, no action was taken (e.g., salts were not applied to the plots by irrigation water). After the winter–spring period, the last 4th period of irrigation period (June–August) was performed, and then irrigation was terminated. After each watering period, soil samples (four replications) were taken from each plot at 0–10, 10–20, and 20–30 cm soil depths.

**Table 1.** Details of irrigation schedule and sampling time.

| Irrigation Number | Date | Soil Moisture % | Applied Water, L | Added Salts by Irrigation Water, g | | | |
|---|---|---|---|---|---|---|---|
| | | | | NaHCO$_3$ | | NaCl | |
| | | | | SAR 20 | SAR 40 | SAR 20 | SAR 40 |
| | | | 1 May 2017—initial sampling | | | | |
| 1 | 17 May 2017 | 25 | 600 | 1.058 | 2.066 | 982.8 | 1.965 |
| 2 | 2 June 2017 | 24 | 630 | 1.11 | 2.167 | 1.032 | 2.065 |
| 3 | 16 June 2017 | 24 | 630 | 1.11 | 2.167 | 1.032 | 2.065 |
| 4 | 18 June 2017 | 34 | 380 | 617 | 1.2 | 573 | 1.15 |
| 5 | 5 July 2017 | 12 | 1040 | 1.835 | 3.58 | 1.7 | 3.408 |
| | | 26 July 2017–28 July 2017—soil sampling (1st period) | | | | | |
| 6 | 29 July 2017 | 20 | 820 | 1.446 | 2.82 | 1.343 | 2.686 |
| 7 | 19 August 2017 | 20 | 820 | 1.446 | 2.82 | 1.343 | 2.686 |
| 8 | 25 August 2017 | 20 | 820 | 1.446 | 2.82 | 1.343 | 2.686 |
| 9 | 28 September 2017 | 23 | 648 | 1.143 | 2.229 | 1.061 | 2.122 |
| 10 | 3 October 2017 | 26 | 576 | 1.016 | 1.981 | 943.4 | 1.886 |
| | | 19 October 2017–22 October 2017—soil sampling (2nd period) | | | | | |
| | | 15 May 2018–17 May 2018—soil sampling (3rd period) | | | | | |
| 11 | 18 May 2018 | 25 | 600 | 1.058 | 2.066 | 982.8 | 1965.6 |
| 12 | 14 July 2018 | 18 | 768 | 1.354 | 2.641 | 1.257 | 2.515 |
| 13 | 20 July 2018 | 17 | 792 | 1.397 | 2.724 | 1.297 | 2.594 |
| 14 | 19 August 2018 | 15.5 | 828 | 1.46 | 2.848 | 1.356 | 2.712 |
| 15 | 26 August 2018 | 16 | 816 | 1.439 | 2.807 | 1.336 | 2.673 |
| | | 18 June 2018–20 June 2018—soil sampling (4th period) | | | | | |

### 2.3. Soil and Water Analysis

Air-dried soil samples were crushed and sieved through a 2 mm sieve and analyzed for texture with the Bouyoucos hydrometer method [37], soil pH and electrical conductivity (EC$_e$) by saturation extract [38], organic matter by modified Walkley Black wet burning method [39], CaCO$_3$ contents with a Scheibler calcimeter [38], and cation exchange capacity (CEC) by treatment with sodium acetate solution (pH 8.2) [40]. The water-soluble Na$^+$, Ca$^{2+}$, and K$^+$ were determined with a flame photometer, Mg with an AAS [41], and CO$_3^{2-}$ and HCO$_3^-$ in saturation extract by titration with 0.01 N H$_2$SO$_4$ and Cl by 0.005 N AgNO$_3$ solution [38]. The SAR was computed by the equation of $\mathrm{Na}/(\sqrt{(\mathrm{Ca}+\mathrm{Mg})/2})$ (Table 2).

**Table 2.** Selected soil and water properties before the experiment.

| Soil Properties | | Soil Depth (cm) | | | Water Quality SAR = 0.98 |
|---|---|---|---|---|---|
| | | 0–10 | 10–20 | 20–30 | |
| Soil texture | | Clay | Clay | Clay | |
| Sand, % | | 27.72 | 28.85 | 30.54 | |
| Clay, % | | 57.88 | 58.34 | 59.26 | |
| Silty, % | | 14.4 | 12.81 | 10.2 | |
| $CaCO_3$, % | | 28.7 | 28.7 | 29.6 | |
| Organic matter, % | | 1.35 | 1.32 | 1.08 | |
| CEC, $cmol_c/kg$ | | 50.7 | 52.5 | 56.1 | |
| pH | | 7.55 | 7.56 | 7.61 | 8.4 |
| $EC_e$, dS m$^{-1}$ | | 0.6 | 0.5 | 0.7 | 0.54 |
| Soluble $Na^+$, meq L$^{-1}$ | | 2.25 | 2.37 | 2.59 | 1.43 |
| Soluble $Ca^{2+}$, meq L$^{-1}$ | | 1.99 | 1.99 | 2.03 | 2.2 |
| Soluble $Mg^{2+}$, meq L$^{-1}$ | | 0.12 | 0.08 | 0.13 | 2.35 |
| Soluble $K^+$, meq L$^{-1}$ | | 0.11 | 0.09 | 0.09 | 0.12 |
| Soluble $Cl^-$, meq L$^{-1}$ | | 2.59 | 2.66 | 2.62 | 1.92 |
| Soluble $HCO_3^-$, meq L$^{-1}$ | | 2.79 | 3.26 | 3.80 | 0.7 |
| Soluble $SO_4^{2-}$, meq L$^{-1}$ | | | | | 1.75 |

$EC_e$: electrical conductivity; CEC: cation exchange capacity.

### 2.4. Statistical Analysis

Data were evaluated using analysis of variance (ANOVA) with the SPSS 20.0. For each water quality and SAR level (NaCl SAR 20; NaCl SAR 40; NaHCO$_3$ SAR 20; NaHCO$_3$ SAR 40) comparison of means ($EC_e$, pH, $Na^+$, $Ca^{2+}$, $Cl^-$, $HCO_3^-$, $Mg^{2+}$, $K^+$) was made at $p < 0.05$ (Figures 2–7; Supplementary Figures S1 and S2). Also, multivariate analysis of variance (multiple ANOVA (MANOVA)) was used to evaluate the contribution of water quality (water type), SAR level, and soil depth and their interactions on the studied soil properties (Tables 3–5). Then, multiple post hoc comparisons (water type, SAR level, and soil depth) were conducted to see where the differences originated. The variation of the mean values of the monitored soil properties by the periods was determined by one-way ANOVA, and the differences between the periods were grouped with 'Tukey' test at a confidence level of $p < 0.05$ (Tables 3–5).

**Table 3.** Multiple comparison tests on the effect of SAR, water type, and soil depth and their interaction with soil properties. Differences are considered significant at the $p < 0.05$.

| Variable | SAR | Water | Depth | SAR × Water | SAR × Depth | Water × Depth | SAR × Water × Depth |
|---|---|---|---|---|---|---|---|
| $Mg^{2+}$ | 0.00 * | 0.05 * | 0.45 | 0.09 | 0.00 * | 0.52 | 0.62 |
| $Na^+$ | 0.00 * | 0.00 * | 0.00 * | 0.00 * | 0.00 * | 0.00 * | 0.00 * |
| $K^+$ | 0.21 | 0.00 * | 0.00 * | 0.01 * | 0.25 | 0.00 * | 0.01 * |
| $Ca^{2+}$ | 0.00 * | 0.00 * | 0.00 * | 0.00 * | 0.00 * | 0.00 * | 0.00 * |
| $HCO_3^-$ | 0.00 * | 0.00 * | 0.00 * | 0.25 | 0.01 * | 0.06 | 0.00 * |
| $Cl^-$ | 0.00 * | 0.00 * | 0.00 * | 0.00 * | 0.00 * | 0.00 * | 0.00 * |
| pH | 0.05 * | 0.00 * | 0.27 | 0.43 | 0.67 | 0.00 * | 0.34 |
| $EC_e$ | 0.06 | 0.00 * | 0.00 * | 0.47 | 0.25 | 0.00 * | 0.91 |

* Within each row, the means labeled with the same letter are not significantly different at $p < 0.05$.

**Table 4.** Effect of SAR level on soil properties.

| Parameter | | Control | Significance | SAR 20 | Significance | SAR 40 | Significance |
|---|---|---|---|---|---|---|---|
| Soluble cations | $Mg^{2+}$ | 0.08ab | 0.22 | 0.12b | 0.18 | 0.05a | 0.22 |
| | $Na^+$ | 5.04a | 1.00 | 16.50b | 1.00 | 23.86c | 1.00 |
| | $K^+$ | 0.09a | 0.31 | 0.10a | 0.31 | 0.11a | 0.31 |
| | $Ca^{2+}$ | 1.70a | 1.00 | 4.76b | 1.00 | 3.94c | 1.00 |
| Soluble anions | $HCO_3^-$ | 4.23a | 1.00 | 6.00b | 1.00 | 5.54c | 1.00 |
| | $Cl^-$ | 3.44a | 1.00 | 16.98b | 1.00 | 22.19c | 1.00 |
| | pH | 7.87a | 0.15 | 7.89ab | 0.15 | 7.96b | 0.19 |
| | $EC_e$ | 0.75b | 1.00 | 2.01a | 0.99 | 2.04a | 0.99 |

Within each row, the means labeled with the same letter are not significantly different at $p < 0.05$.

**Table 5.** Effect of the irrigation water type on soil properties.

| Parameter | | Control | Significance | NaCl | Significance | NaHCO$_3$ | Significance |
|---|---|---|---|---|---|---|---|
| Soluble cations | Mg$^{2+}$ | 0.08a | 0.20 | 0.10a | 0.20 | 0.07a | 0.20 |
| | Na$^+$ | 5.44a | 1.00 | 28.96c | 1.00 | 11.40b | 1.00 |
| | K$^+$ | 0.03a | 0.27 | 0.14b | 1.00 | 0.07a | 0.27 |
| | Ca$^{2+}$ | 1.70a | 1.00 | 5.33c | 1.00 | 1.37b | 1.00 |
| Soluble anions | HCO$_3^-$ | 4.23a | 1.00 | 4.38b | 1.00 | 6.96c | 1.00 |
| | Cl$^-$ | 3.44a | 1.00 | 33.77c | 1.00 | 5.40b | 1.00 |
| | pH | 7.87a | 0.08 | 7.75a | 0.08 | 8.09b | 1.00 |
| | EC$_e$ | 0.75b | 0.53 | 3.08a | 1.00 | 1.11b | 0.53 |

Within each row, the means labeled with the same letter are not significantly different at $p < 0.05$.

## 3. Results

### 3.1. Soil Properties before the Experiment

The initial field soil properties (before irrigation with sodic water) and the fresh irrigation water quality parameters are provided in Table 2. The soil is generally characterized by clay texture, slight alkalinity, low organic matter, and high CaCO$_3$ content at all depths. The variation in clay, pH, EC, and CaCO$_3$ and most of the soluble cation's content was mainly lower than the variation in organic matter and cation exchange capacity (CEC). Non-saline irrigation waters with very low sodicity had an alkaline reaction.

### 3.2. Effect of Irrigation on Water-Soluble Ion Concentration

The results of the change in soil EC$_e$ in the soil profile are provided in Figure 2. The EC$_e$ values increased in the soil surface (primarily) and sub-surface layer during the 1st, 2nd, and 4th periods of irrigation, especially by NaCl water applications (from 0.6 to 5.5 dS m$^{-1}$) rather than NaHCO$_3$ water treatment (from 0.6 to 1.7 dS m$^{-1}$). During the 3rd period, salts leached out, and EC$_e$ decreased intently to the initial values (e.g., NaCl: 1.0–1.3 dS m$^{-1}$; NaHCO$_3$: 0.8–1.0 dS m$^{-1}$) or moved from the upper to the lower soil layers and accumulated there. Remarkably, both NaCI and NaHCO$_3$ water treatments significantly contributed to soil EC$_e$ at the SAR 40, particularly at the upper layer (0–10 cm). However, the changes in EC$_e$ values were far less dynamic in the soil profile with the application of NaHCO$_3$ water (Figure 2).

Soil pH was 7.55 at the beginning and decreased after the 1st and 2nd periods of NaCl water applications but exceeded 8.0 and 7.7 after the 3rd and 4th periods of irrigation for two SAR levels, respectively. Although the increments after 3rd period were significant, the effect of NaCl water on soil pH was insignificant for the entire experiment period. In NaHCO$_3$ water applications, the soil pH decreased insignificantly after the 2nd period of irrigation and then increased (to 7.9–8.3) after 3rd and 4th periods of irrigation, particularly in the top layer (0–10 cm). There was a significant increase in soil pH for both SAR levels, especially for the SAR 40, and soil pH reached 8.3 in the top layer after the 4th irrigation period (Figure 3). For both water quality, water-soluble Na$^+$ increased in the 1st, 2nd, and 4th periods in both the surface (from 2.4 to 53.5 meq L$^{-1}$) and subsurface (~23 meq L$^{-1}$) layers; in comparison with NaHCO$_3$, the NaCl water applications yielded two-times higher Na$^+$. The increases between sampling periods were significant. There was no clear trend in Na$^+$ distribution in the soil profile, yet Na$^+$ concentration was higher, mainly in the top layer (Figure 4).

Under irrigation, in general, water-soluble Na$^+$ decreased the water-soluble Ca$^{2+}$ concentration in the top layer of soil; water-soluble Ca$^{2+}$ ions were leached from the top layer (0–10 cm) and accumulated at a depth of 10–20 cm, and depending on the water quality, some changes were also noticed in the 20–30 cm depth. Compared to the initial control value (1.5 meq L$^{-1}$), water-soluble Ca$^{2+}$ increased at the SAR 20 of NaCl water applications in the 1st, 2nd and 4th periods (4.7–7.0 meq L$^{-1}$) in the top layer, whereas for the SAR 40 of NaCl treatment, Ca$^{2+}$ decreased considerably after the 2nd or 3rd period of irrigation (0.9–1.2 meq L$^{-1}$), except for the 4th period, when Ca$^{2+}$ increased to 12.2 meq L$^{-1}$

at the end of irrigation. It should be noted that after winter–spring precipitation (3rd period), concentrations of $Ca^{2+}$ were similar to the initial values (Figure 5). Excess soil Na concentration in the NaCl SAR 40 application could be related to a rapid transport of soluble $Ca^{2+}$ from the top to deeper soil layers.

In both SAR levels of $NaHCO_3$ water applications, water-soluble $Ca^{2+}$ in the soil decreased after the 1st and 2nd irrigation periods, while the trend in different depths varied notably (Figure 5). Although there was a slight increase in the 3rd period (0–20 cm) compared to the first two periods, it reached a value somewhat close to the initial value of water-soluble $Ca^{2+}$ after the 4th period. For the $NaHCO_3$ water use, the differences in soil water-soluble $Ca^{2+}$ between the periods were significant (Figure 5).

The water-soluble $Mg^{2+}$ in the soils showed irregular increases and decreases in the SAR 20 of NaCl water application but later displayed an apparent reduction in the 3rd period and a net increase in the 4th period of irrigation. After the 1st period, the water-soluble $Mg^{2+}$ at both SAR levels of $NaHCO_3$ water use decreased compared to the initial concentration at all depths. At the same time, it remained almost non-existent in the 2nd period. $NaHCO_3$ water seemingly affected the water-soluble $Mg^{2+}$ concentration in the soil, even at depths of 20–30 cm. Similar results were obtained at SAR 20 and SAR 40 in the 3rd and 4th periods, revealing that the water-soluble Mg increased slightly compared to the 2nd period. The changes in water-soluble $Mg^{2+}$ in the $NaHCO_3$ water treatments between the periods were not statistically significant (Supplementary Figure S1).

The water-soluble $K^+$ first decreased with NaCl water applications at both SAR levels and increased to the control value with continuous irrigation. During the 3rd period, it fell with winter–spring rains and rose again in the 4th period, especially in the top layer. While the increase in the 4th period was significant, the changes in other periods were insignificant. At both SAR levels of $NaHCO_3$ water use, water-soluble $K^+$ decreased in the 1st, 2nd, and 3rd periods compared to the initial control value; however, during the 4th period, it increased, especially in the surface layer. While the changes in the 1st, 2nd, and 3rd periods for SAR 20 treatments were significant, the differences were insignificant for SAR 40 (Supplementary Figure S2).

At both SAR levels of NaCl water applications, water-soluble $Cl^-$ in the surface soil layer increased rapidly after the 1st and 2nd periods; this increase was more noticed in SAR 40 (38.0 meq $L^{-1}$) water than in SAR 20 (23.2 meq $L^{-1}$) water. In the 3rd period, after the winter–spring precipitation, the water-soluble $Cl^-$ in the soil decreased rapidly and increased again during the 4th period (~50 meq $L^{-1}$). The changes between the periods were significant (Figure 6). For the $NaHCO_3$ water applications, however, water-soluble $Cl^-$ in the soil surface was not increased significantly at both SAR levels until the end of the 2nd period. Values similar to the initial $Cl^-$ concentrations were noticed in the 3rd period; however, in the 4th period, $Cl^-$ increased significantly for both SAR levels (Figure 6).

The water-soluble $HCO_3^-$ did not change until the end of the 1st and 2nd periods in the top layer at both SAR levels of the NaCl water applications. During the winter–spring precipitation, water-soluble $HCO_3^-$ increased in the 3rd period (from ~3 to 7 and 11.7 meq $L^{-1}$) and decreased again in the 4th period; the changes between the 3rd and 4th periods were significant. The $NaHCO_3$ water use did not significantly change water-soluble $HCO_3^-$ in the soil until the 2nd period for all depths (at the SAR 20). However, after the 2nd period, the water-soluble $HCO_3^-$ in the soil decreased and again increased (8.5 meq $L^{-1}$) significantly in the 4th period of irrigation. Similar to the NaCl application, these differences between the 3rd and 4th periods were significant (Figure 7).

### 3.3. Multivariate Analysis of Variance

The effects of SAR level, water type, and soil depth on soil properties (except $K^+$, $Mg^{2+}$, or pH) were significant (Table 3). Thus, the effects of SAR level, water type, and soil depth and their interactions on studied soil properties were analyzed (Tables 4–6). The interaction effects of the factors on soil properties were changeable: (i) SAR level x water type effect was significant for $Na^+$, $Ca^{2+}$, $K^+$, and $Cl^-$; (ii) SAR level × soil depth effect was weighty

for $Na^+$, $Ca^{2+}$, $Mg^{2+}$, $HCO_3^-$, and $Cl^-$; (iii) water type × soil depth effect was important for all properties, except $Mg^{2+}$; and (iv) triple interaction effect was significant for $Ca^{2+}$, $Na^+$, $Mg^{2+}$, $Cl^-$, and $HCO_3^-$, i.e., for most of the ions (Table 3). While the influence of the SAR levels of irrigation waters on the mean concentration of water-soluble ions ($Na^+$, $Ca^{2+}$, $Cl^-$ and $HCO_3^-$) and soil $EC_e$ were diverse, the difference between them (control SAR 1 and SAR 20 and SAR 40) were significant (Table 4). The contributions of water types (control fresh, NaCl, and $NaHCO_3$) were significant for water-soluble $Na^+$, $Ca^{2+}$, $Cl^-$, and $HCO_3^-$, and also for $EC_e$ (NaCl) and $pH_e$ ($NaHCO_3$) (Table 5). Though the effect of soil depth was significant for water-soluble ions ($Na^+$ and $Ca^{2+}$, $K^+$, $HCO_3^-$, $Cl^-$) and $EC_e$, the significant difference was found at 0–10 cm depth with the highest mean of studied properties (Tables 3 and 6).

**Table 6.** Effect of irrigation on soil properties at various depths.

| | Parameter | 0–10 cm | Significance | 10–20 cm | Significance | 20–30 cm | Significance |
|---|---|---|---|---|---|---|---|
| | $Mg^{2+}$ | 0.09a | 0.83 | 0.08a | 0.83 | 0.09a | 0.83 |
| Soluble | $Na^+$ | 21.60c | 1.00 | 13.71a | 1.00 | 16.15b | 1.00 |
| cations | $K^+$ | 0.16a | 1.00 | 0.07b | 0.87 | 0.07b | 0.87 |
| | $Ca^{2+}$ | 4.83a | 1.00 | 2.11b | 1.00 | 2.11b | 1.00 |
| | $HCO_3^-$ | 5.93a | 1.00 | 5.35b | 0.12 | 5.11b | 0.12 |
| Soluble | $Cl^-$ | 23.13a | 1.00 | 12.74b | 0.68 | 13.19c | 0.68 |
| anions | pH | 7.94a | 0.19 | 7.92a | 0.19 | 7.88a | 0.19 |
| | $EC_e$ | 2.69a | 1.00 | 1.26b | 0.91 | 1.38b | 0.91 |

Within each row, the means labeled with the same letter are not significantly different at $p < 0.05$.

## 4. Discussion

Management practices, local meteorological events (temperature, wind, radiation, and air humidity), and soil conditions (soil texture, salinity, lime content, water content, and tillage practices) significantly affect soluble ion movement and distributions in the soil profile. The permeability of clay soil was low, affecting both the velocity of water and ion transport and the kinetics of ion exchange processes between solid and solution phases [25]. In the 1st, 2nd, and 4th periods of irrigation, $EC_e$ values on the soil surface and subsurface layers increased significantly in terms of SAR levels for both sodic water types, mostly when NaCl water was used (Figure 2). The main reasons for these differences were related to (i) the continuous addition of salts to the soil with sodic irrigation water (Table 1), (ii) the variation in the solubility of NaCl and $NaHCO_3$ salts in the presence of high soil $CaCO_3$, and (iii) the variation in the mobility and accumulation rate of ions at different depths in the soil profile, which was in line with the previous studies [42–45]. The salts accumulated during the first irrigation season (1st and 2nd periods) were leached out by the precipitation (from the surface towards the deeper horizons) in the 3rd period, leading to a significant decrease in soil $EC_e$ (Figure 2). Since most of the precipitation in the study area falls in the winter–spring seasons, it was established that most of the salts in the clay soil profile accumulated within the upper 0–30 cm layer and a small amount (<30%) was transported into the deeper layers. However, in the 3rd watering period, a considerable downward movement of various ions depended on their solubility and chemical properties. Many studies from arid and semi-arid regions found that the salt content of the clay soil increases during the irrigation season and decreases in the rainy season due to leaching [46–48]. While the ions in the soil profile are transported downwards with irrigation or rain, they move upwards by capillary forces during evaporation, particularly in clayey soils and when water with high sodium is used. The increase in $EC_e$ by irrigation that was observed in the 4th period and the upward movement of water with capillary forces were the main reasons for the rise of EC again to the soil surface [49]. Öztürk and Özkan [49] have reported that soil $EC_e$ at the soil surface after ten days of evaporation period increased to up to 31% and 46% of the pre-drying level for the clay loam and sandy clay loam soils, respectively.

This study revealed that NaCl water application did not significantly change the soil pH due to its neutral salt characteristics (Figure 3). The buffering property of the soil

and the further release of $Ca^{2+}$ ions from $CaCO_3$ would have also contributed to those processes. Sreenivas [50] reported a negative relationship between the soil EC and the pH of the saturation extract; the author noted that the high NaCl salt concentration in the irrigation water prevented an increase in the soil pH. Several studies from dry regions also have shown that the salinity of irrigation water and the intervals between them do not affect soil pH [51–53]. On the other hand, Pessoa et al. [48] found that in salt-affected sandy and silty loamy soils, irrigation water with varying types of salt containing $Cl^-$ increased the rate of the soluble $Cl^-$ over $CO_3^{2-}$ and $HCO_3^-$ concentration in the soil; thus, pH in the soil extract decreased. In the current study, the increases in the pH values after the 3rd and 4th periods in the SAR 40 level of the $NaHCO_3$ water were significant compared to the other periods. This change, especially in $NaHCO_3$ applications in the 3rd period, was related to the hydrolyses of exchangeable $Na^+$ into NaOH, leaching down with winter–spring precipitation, and the formation of $Na_2CO_3$ from NaOH reacting with $CO_2$, absorbed from the air or produced by microorganisms, and the high content of the bicarbonates in the irrigation water [54]. Saygın et al. [54] explained that a slow but steady increase in calcareous soil pH value at the end of two growing seasons with $NaHCO_3$ irrigation waters with a SAR 20 is linked to the buffering effect of $CaCO_3$ and the formation of NaOH. Numerous studies reported that elevated $HCO_3^-$ in the irrigation water caused a rise in soil pH under a combination of rain and irrigation or irrigation alone [55,56].

With the sodic water application, the water-soluble $Na^+$ content increased together with the $EC_e$ of soil, and subsequently, $Na^+$ leached downwards, especially when NaCl water was used (Figure 4). Researchers from different countries have reported a highly positive relationship between soil $EC_e$ and water-soluble $Na^+$ [48,57–60]. In a study conducted for two years, the $Na^+$ concentration increased up to 21.6% for three different salt levels in irrigation water, and $Na^+$ concentration was consistent with soil $EC_e$ [61]. In our study, although a similar effect was evident for NaCl water applications, these changes were not observed for $NaHCO_3$ water type. Salts with high dissolution rates can immediately mix into the soil solution and cause the effects of these salts (such as NaCl) on soil salinity to differ from those of salts with low solubility in water [62].

The presence of a high amount of $Ca^{2+}$ in the soil solution was related to the high Na+ ion concentration after the treatment of sodic irrigation water (Figures 3 and 4). The soil became salinized during the SAR 20 of NaCl water application, yet it was not fully saturated with $Na^{2+}$ at the end of the 1st or 2nd periods. At the beginning of irrigation, $Ca^{2+}$ was the major cation in the soil solid and solution phase due to its strong bonding force, and it was associated with calcareous soil characteristics. In the SAR 40 of NaCl water application, the soil became more saturated with $Na^+$ due to its increased concentration, which tended to increase soil pH and decrease the concentration of water-soluble $Ca^{2+}$ in the surface layer compared to the initial control one; subsequently, $Ca^{2+}$ moved to the lower depths due to the displacement of $Na^+$ with $Ca^{2+}$ in the soil absorption sides. It was found that $Ca^{2+}$ released from the exchange complexes with NaCl applications before the rainy season (rainfall with no electrolytes) was leached downwards only during the 3rd period and thus was removed from the soil surface layer. Sodic water application in the 4th period increased the water-soluble $Ca^{2+}$ concentration at the soil surface since the $Ca^{2+}$ on the colloid surfaces did not have a chance to be leached out yet. In addition, the low soluble $CaCO_3$ reacted with sodium, dissolved some of its $Ca^{2+}$ ions, and released them into the soil solution environment. The excess amount of $Na^+$ was replaced by $Ca^{2+}$, which was bound to the adhesion surfaces and caused the release of $Ca^{2+}$ ions into the soil solution and, finally, increased $Ca^{2+}$ concentration in the leachates [26,54].

For $NaHCO_3$ treatment, the concentration of the water-soluble $Ca^{2+}$ and $Mg^{2+}$ in the soil solution decreased at all depths for SAR 20 and SAR 40 applications (Supplementary Figure S1) due to the increased concentration of water-soluble $Na^+$ in semi-arid soils treated with sodic water; consequently, the release of $Ca^{2+}$ and $Mg^{2+}$ into the solution led to the decrease in their concentrations in the surface layers [63,64]. Analogous results were found in this study: the decrease in the water-soluble $Mg^{2+}$ concentration, along with the salinity

level and composition of the irrigation water, was associated with the increase of the $Na^+$ concentration in the soil solution with the sodic irrigation waters. Tavakkoli et al. [65] reported that treating sandy soils with NaCl waters decreased the exchangeable $Mg^{2+}$ concentration in the semi-arid region of South Australia. Moreover, Pessoa et al. [48] and Saygın et al. [53] established that using irrigation waters with different EC and SAR values decreased the exchangeable $Mg^{2+}$ concentration similar to the $Na^+$–$Ca^{2+}$ shift in soil exchange complexes. The saline irrigation water did not significantly affect the change of water-soluble $K^+$ values at the soil surface and the subsurface layers (Supplementary Figure S2). The reason could be related to the low $K^+$ concentration in irrigation water and soil, as the high $Na^+$ concentration in the water quickly washed away $K^+$ from the adsorptive surfaces. Generally, $K^+$ and $P^{3+}$ were found in low concentrations in soil solutions [66].

Depending on the SAR level, the water-soluble $Cl^-$ concentration in the soil increased by NaCl water applications during the first irrigation seasons and decreased in the 3rd period when $Cl^-$ was leached out of the soil surface with rainfall with no salts (Figure 6). On the contrary, $NaHCO_3$ water applications did not significantly affect the water-soluble $Cl^-$ concentration in the soil, except for the 4th period. It could be explained by the high $Na^+$ and $Cl^-$ concentrations in the water and the rapid dissolution rate of these ion compounds. There is a synergistic relationship between $Na^+$ and $Cl^-$. Previous studies have also shown that $Cl^-$ is generally the most active ion in solute transport processes, followed by sulfate and relatively stagnant carbonate (bicarbonate) [67]. A very low $Cl^-$ concentration in the top layer after the irrigation and a sharp increase in $Cl^-$ concentration after the evaporation period was explained by the high mobility of this ion [68]. Pondkule et al. [56] found that $Cl^-$ in the soil extract before irrigation varied between 4.8 and 7.2 meq $L^{-1}$, while it raised to 5.2 and 8.80 meq $L^{-1}$ after irrigation. Soil texture also affects the retention of these ions in the soil. Due to their high water-holding capacity, $Na^+$ and $Cl^-$ accumulations in fine-textured soils were higher than in coarse-textured soils [48]. In the $NaHCO_3$ water treatments, the high soluble $Cl^-$ concentration at the soil top layer during the 4th period may be related to the moving of $Cl^-$ ions, which leached and accumulated at specific depths during the rainy season, or to the soil surface by evaporation [59].

In the NaCl water treatment, the amount of $HCO_3^-$ in the soil increased after the 3rd period. This could be explained by the increased solubility of $HCO_3^-$, which has low solubility with NaCl salt (Figure 7). At the same time, it was observed that water-soluble $HCO_3^-$ increased at SAR 40 in the 4th period and accumulated at lower depths due to the precipitation of $Ca^{2+}$. However, it should be noted that $NaHCO_3$ water applications (at SAR 40) increased the soil $HCO_3^-$ concentration continuously, although, in the 3rd period, the concentration of $HCO_3^-$ ions was expected to decrease. The difference in leaching rate or $HCO_3^-$ ion distribution in soil profile can also be explained by the fact that saturated hydraulic conductivity of clay soil strongly depends on both EC of water and water salinity type [10,21]. Application of four pore volumes of $NaHCO_3$ sodic water (EC = 3 dS $m^{-1}$, SAR = 20) in similar clay soils decreased saturated hydraulic conductivity from ~0.4 cm $min^{-1}$ to 0.15 cm $min^{-1}$ [27,37]. Because of its low solubility and mobility characteristics in clay soils, winter–spring precipitation could not have leached $HCO_3^-$ from the top layers of semi-arid soils, allowing the carbonates to accumulate in the soil profile [69]. As reported [70], the soil $HCO_3^-$ concentration was strongly affected by irrigation water quality, soil depth, and sampling time. Pondkule et al. [56] stated that the $HCO_3^-$ concentration in the soil extract increased from 7.3 to 8.4 meq $L^{-1}$ to 7.5 to 8.9 meq $L^{-1}$ at the end of the irrigation with sodic water. When the water-soluble $HCO_3^-$ concentration was high, it reacted with the $Ca^{2+}$ in the soil and precipitated as $CaCO_3$. This condition led to an increase in the $Na^+$ soil sodicity ratio in the soil [71]. In another study, a sharp decrease in $NO_3^-$, $SO_4^{2-}$, and $Cl^-$ concentrations, but not in $HCO_3^-$, at the beginning of the leaching and, later, steady-state concentrations were observed in the collected leachates when irrigation water with $NaHCO_3$ was used [54].

## 5. Conclusions

Calcium carbonate, which is generally found in the soil profile in such semi-arid areas or dry regions, plays a vital role in reducing or delaying the effects of possible problems associated with sodic water application and soil chemical and physical quality deterioration. Increasing $Na^+$ in irrigation water increased the solubility of $Ca^{2+}$ in the form of $CaCO_3$, which has low solubility, and enhanced its leaching to lower soil depths. The released $Ca^{2+}$ interacted with other cations in the soil, causing further positive physicochemical changes in the soil solution, yet the kinetics of the processes were dependent on water type. Although the amount of water-soluble $Ca^{2+}$ released through ion exchange phenomena in soil exchange complexes decreased for the short-term period with sodium irrigation water application, the excess lime content in the soil can guarantee $Ca^{2+}$ restoration in the long term. Results show that physical and chemical deteriorations that may occur in the soil can be prevented or limited to a certain extent, particularly when sodic water is used to balance the deficit of irrigation water. Therefore, the results of this study can be used for sustainable irrigation management plans in arid and semi-arid regions, considering site-specific conditions associated with soil properties and available irrigation water quality. Increasing water productivity and making safe use of poor-quality water in agriculture will play a vital role in easing competition for scarce water resources, preventing environmental degradation, and providing food security.

**Supplementary Materials:** The following supporting information can be downloaded at: https://www.mdpi.com/article/10.3390/soilsystems8010010/s1, Figure S1: Water-soluble $Mg^{2+}$ of soil as a function of irrigation water applications during the irrigation periods (1, 2, and 4) and the rainy period (3) for four water qualities: (a) NaCl water: SAR 20; (b) NaCl water: SAR 40; (c) $NaHCO_3$ water: SAR 20; and (d) $NaHCO_3$ water: SAR 40. For each water quality, within each soil depth, the columns labeled with the same letter; Figure S2: Water-soluble K+ of soil as a function of irrigation water applications during the irrigation periods (1, 2, and 4) and the rainy period (3) for four water qualities: (a) NaCl water: SAR 20; (b) NaCl water: SAR 40; (c) $NaHCO_3$ water: SAR 20; and (d) $NaHCO_3$ water: SAR 40. For each water quality, within each soil depth, the columns labeled with the same letter.

**Author Contributions:** Conceptualization, A.E.P. and H.S.Ö.; methodology, A.E.P. and H.S.Ö.; investigation, A.E.P.; field experiments, A.E.P.; writing—original draft preparation, A.E.P., H.S.Ö. and A.I.M.; writing—review and editing, A.E.P., H.S.Ö. and A.I.M.; visualization, A.E.P. and A.I.M. All authors have read and agreed to the published version of the manuscript.

**Funding:** This research received no external funding. The authors thank Ankara University for the financial support.

**Institutional Review Board Statement:** Not applicable.

**Informed Consent Statement:** Not applicable.

**Data Availability Statement:** The raw data supporting the conclusions of this article will be made available by the authors on request.

**Conflicts of Interest:** The authors declare no conflicts of interest.

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
