# Peer review of "The Effect of Sodic Water Type on the Chemical Properties of Calcareous Soil in Semi-Arid Irrigated Land"

_soilsystems, doi:10.3390/soilsystems8010010_

Round 1
Reviewer 1 Report
Comments and Suggestions for Authors
This paper aimed to evaluate the effect of sodic water type (quality) on the chemical properties of a calcareous clay soil in semi-arid irrigated land to monitor the potential of medium-term changes (change in water-soluble ion concentration and ECe) in a calcareous clay soil irrigated with two types of sodic water (NaCl 445 and NaHCO3; SAR 20 and SAR 40). This topic is interesting, and the paper is well structured and well written. But, some comments need to be solved. It is recommended that minor revisions to address these concern.
1. Line 110-111: “The initial soil properties were used as the control one to evaluate contribution of irrigation with sodic water quality on soil properties.” What are the soil properties and which soil property parameters have been compared and analyzed? Here should be specific points. Although Figures 1-6 have already shown.
2. The font and image annotations in Figures 1-6 should be consistent.
3. Line 64-65: “As a result, reduction in aeration and permeability in the soil were noted due to soil aggregate slaking and clay dispersion and clogging soil pores.” Some references should be cited to prove the feasibility of this statement, such as Journal of Hydrology, 2023, 618, 129230; Journal of Hydrology, 2022,614, 128583...
Author Response
Dear Reviewer
We are submitting a revised version of our manuscript soilsystems-2630097 “Effect of sodic water type on chemical properties of a calcareous soil in semi-arid irrigated land” to be considered for publication in soil systems.
We would like to thank you for the valuable comments and directions. We responded your comments as fallows;
Reviewer # 1 comments (R1)
This paper aimed to evaluate the effect of sodic water type (quality) on the chemical properties of calcareous clay soil in semi-arid irrigated land to monitor the potential of medium-term changes (change in water-soluble ion concentration and ECe) in a calcareous clay soil irrigated with two types of sodic water (NaCl 445 and NaHCO3; SAR 20 and SAR 40). This topic is interesting, and the paper is well structured and well written. But some comments need to be solved. It is recommended that minor revisions to address this concern.
Response: Thank you very much for your comment.
R1: Line 110-111: The initial soil properties were used as the control one to evaluate contribution of irrigation with sodic water quality on soil properties. What are the soil properties and which soil property parameters have been compared and analyzed? Here should be specific points. Although Figures 1-6 have already shown.
Response: The soil properties used for comparisons were added to the revised version. Also, Figures 1-6 were revised.
R1: The font and image annotations in Figures 1-6 should be consistent.
Response: The font and annotations were edited in all the Figures (1-6) to make them consistent
R1: Line 64-65: “As a result, reduction in aeration and permeability in the soil were noted due to soil aggregate slaking and clay dispersion and clogging soil pores.” Some references should be cited to prove the feasibility of this statement, such as Journal of Hydrology, 2023, 618, 129230; Journal of Hydrology, 2022,614, 12858
Response: The following references were added to the revised version as advised.
Wen, T; Chen, X; Shao, L. Effect of multiple wetting and drying cycles on the macropore structure of granite residual soil. Journal of Hydrology. 2022, Volume 614, Part B, 128583, ISSN 0022-1694, https://doi.org/10.1016/j.jhydrol.2022.128583.
Wen, T; Chen, X; Luo, Y; Shao, L; Niu, G. Three-dimensional pore structure characteristics of granite residual soil and their relationship with hydraulic properties under different particle gradation by X-ray computed tomography. Journal of Hydrology. 2023, Volume 618, 129230, ISSN 0022-1694, https://doi.org/10.1016/j.jhydrol.2023.129230.
Reviewer 2 Report
Comments and Suggestions for Authors
Dear Authors!
I think, Your research deal with a very important aspect of irrigation problems. It is a well-written work of You which based on well designed experiments. The results are correct and discussoion subsection builds on that results comparing them to finding of other researches. However, I think that You should interpret Your methods more clearly, because there were some obscure points for me. I marked that parts in the attached document. Also, in my opinion sometimes You use too much citations in one sentence.

Author Response
Dear Reviewer
We are submitting a revised version of our manuscript soilsystems-2630097 “Effect of sodic water type on chemical properties of a calcareous soil in semi-arid irrigated land” to be considered for publication in soil systems.
We would like to thank you for the valuable comments and directions. We responded your comments as fallows;
Reviewer # 2 comments (R2)
Dear Authors!
I think, Your research deals with a very important aspect of irrigation problems. It is a well-written work of You which based on well designed experiments. The results are correct and discussoion subsection builds on that results comparing them to finding of other researches. However, I think that You should interpret Your methods more clearly, because there were some obscure points for me. I marked that parts in the attached document. Also, in my opinion sometimes You use too much citations in one sentence.
Response: Thank you very much for your comment.
R2: You should interpret Your methods more clearly because there were some obscure points for me. I marked that parts in the attached document
Response: Some corrections and additions were done in Material and methods to make it clear. We also considered your marks on the manuscript, corrected them and replied to each comment of you. A map showing the location and topography of the experimental area was added.
R2: You use too much citations in one sentence.
Response: We agree that too many citations were used in the paper. We removed a considerable amount of them in the revised version (from 85 to 71).
R2: L.38. I don't think that irrigation itself can double the productivity of agriculture land. Maybe we should take into other factors (the amount of applied fertilizers for example)
Response: We checked the manuscript. Authors concluded this statement for 'arid conditions'. Yet the advised change was made.
R2: L.51. Please give a short explanation why the people use saline-sodic water to irrigate arable lands, if they know that cause soil degradation.
Response: An explanation is added to the revised version.
R2: L. 62. Too much citations, maybe less would be enough.
- 67. also too much citations refer to only one sentence
- 79. also too much citations
- 312 too much citations
L.354. too much citations
Response: We agree. Several of references (relatively less relevant or similar ones) were removed
R2: L.84. What about the usual quality of irrigation waters in Turkey? Do these waters contain a high amount of salines or carbonates? are these waters appropriate for irrigation in the terms of their quality?
Response: The quality of available water in the region (semi-arid and arid region), including study area mostly is not suitable for irrigation and mostly contains significant amounts of sodium. Thus, we designed this study to evaluate the possible effects of sodium on soil properties to simulate the condition. An explanation is added to the revised version of the manuscript.
R2: L. 89. Why did You choose that soil type? Are they important soil types in agricultural production in Turkey?
Response: Clay soil is considered more complicated (physical and chemical deterioration). This soil was explicitly chosen to prove to what extent clay soil with high calcium carbonate can tolerate the adverse effects of sodic water irrigation. An explanation is added to the revised version.
R2: L. 97. A map, which contain the main topographical features of the study site would be useful.
Response: A map showing the location and topography of the experimental area was added.
R2: L. 103. How many field plot did You use? What is the total area of the experimental site?
Response: In a total of 15 parcels/plots (5 water quality x 3 replications) were used. Water quality is related to control (non-salinized good quality water) and 4 sodium water qualities. An explanation is added to '2.2 Experimental design' section.
R2: L.106. What is the exact time of these periods? In Table 1., I only see the time of soi sampling, which took only two days.
Response: Table 1 shows the time of irrigation and soil sampling together. The first column shows irrigation number and second column shows the time of irrigation. Soil samples were collected after five irrigations (five irrigation = 1 period).
R2: L.111. Were controls site (where was no irrigation at all) used in the experiments? Also, were such an experimental site used where only good quality fresh water was used? It could also prove the chance of comparison of changed soil properties.
Response: The initial soil properties were determined at the control parcels (plots) at the beginning of the experiment. In the Figures 1-5 we mainly compared the chemical properties of the soils treated with control (good quality water) and the sodic water after every 5 irrigations (period). In other words, yes, we used a good quality water in the experiment: we prepared 4 sodic-saline water and used a non-salinized good quality water (a total of 5 water quality) to compare the results.
R2: L. 119, Did 276,2 mm precipitation fall in this period?
Response: Yes, it occurred: 276,2 mm rain fell in the 3rd period. The amounts of the precipitation during each period are presented in LL. 108 -110.
R2: Table 2. Every experimental field plots have the same soil properties or are any differences between them?
Response: The experimental area is about 400 m2, and considered as very homogeneous. It appears also from our previous studies. However due to the crack formation throughout the soil properties may shows slight variety. Since we irrigated the soil regularly, cracks were not formed properly, and soil showed a homogeneous property.
R2: Figure 1. Please use markers on y-axis between number values in every bar graph.
Response: The statistics, abbreviations and format are re-written as advised in the revised version.
R2: L. 321. I thin k that it should be true if people use water with high amount of sodic materials as irrigation water
Response: True. Added.
R2: L. 326. That is a self-citation
Response: Yes, it is one of two papers of authors used in this manuscript. We have several papers on soil salinity. However, we used two of them (more relevant) in this paper.
R2: L. 439-440. this statement suits into Introduction section.
Response: This part moved to the Introduction section in the revised version.
R2: L.444-446. That is method, not conclusion.
Response: This sentence was deleted
R2: L.449. That is result
Response: Correct. However, this is one of the most important outputs of the study. We used this information for the general part of the conclusion section to indicate the contribution of the physicochemical mechanisms in calcareous soils.

Reviewer 3 Report
Comments and Suggestions for Authors
The manuscript deals with the effect of sodic water addition on soil properties in semi-arid irrigated land. The topis of manuscript is centairly actual, however the manuscript should be re-written to improve its potential for readers.
Authors should better explain the aim of the study. It is not clear why they are dealing with this soil type and why they are focused on sodic water type.
Could authors more discuss the effect on other soil constituents (quaility of organic matter, calcium and magnesium content).
Did authors monitored the effect of wheather, mainly storms, rain etc. on potential leaching and flushing of salts and other soil constituents? and for the dry season?
Why only 4 irrigation periods were realized?
In my opinion, authors should include the above-mentioned recommendations into the text. It isalso necessary to re-writte the introduction part to explain the aim odf their study.
Comments on the Quality of English LanguageStandard level.
Author Response
Dear Reviewer
We are submitting a revised version of our manuscript soilsystems-2630097 “Effect of sodic water type on chemical properties of a calcareous soil in semi-arid irrigated land” to be considered for publication in soil systems.
We would like to thank you for the valuable comments and directions. We responded your comments as fallows;
Reviewer # 3 comment (R3)
The manuscript deals with the effect of sodic water addition on soil properties in semi-arid irrigated land. The topic of the manuscript is centairly actual; however, the manuscript should be re-written to improve its potential for readers.
Response: Thank you very much for your comment. All the comments were addressed
R3: Authors should better explain the aim of the study. It is not clear why they are dealing with this soil type and why they are focused on sodic water type.
Response: This part was revised. We explained why a calcareous clay soil were selected for this study.
R3: Could authors more discuss the effect on other soil constituents (quality of organic matter, calcium and magnesium content).
Response: Some soil constituents were provided and discussed in ‘2.1. Location’
R3: Did authors monitored the effect of weather, mainly storms, rain etc. on potential leaching and flushing of salts and other soil constituents? and for the dry season?
Response: Since the experimental area is in an arid environment, extreme rain storms are not expected in the dry season. We did not monitor each week the ion movements during the experiment, but after 5 irrigation (one period), i.e., certain anions and cations (e.g., Ca, Mg, K, HCO3, Cl, EC) and their consequences were analysed and evaluated for every five irrigations. Consequently, the results are discussed in several places in the manuscript.
R3: Why only 4 irrigation periods were realized?
Response: We expected that degradation caused by sodic waters in a calcareous soil would initially occur on or before 4 - 6 periods, each with five irrigation activities. According to the available preliminary soil physical properties (e.g., aggregate stability, bulk weight and infiltration tests), the deterioration started within the third period and became evident in the fourth period. Therefore, we decided that four periods would be sufficient.
Reviewer 4 Report
Comments and Suggestions for Authors
The statistics shown on figures are confusing.
For example, ab bars higher than a, b bars higher than a, bc higher than ab, ab higher than a, b higher than a, c higher than b.
Need to correct that or provide explanation.
I stopped at the figures. I could not continue reviewing the manuscript with the concern about statistics.
Author Response
Dear Reviewer
We are submitting a revised version of our manuscript soilsystems-2630097 “Effect of sodic water type on chemical properties of a calcareous soil in semi-arid irrigated land” to be considered for publication in soil systems.
We would like to thank you for the valuable comments and directions. We responded your comments as fallows;
Reviewer # 4 comments (R4)
Response: Thank you very much for your comment
R4: The statistics shown on figures are confusing. For example, ab bars higher than a, b bars higher than a, bc higher than ab, ab higher than a, b higher than a, c higher than b.
Need to correct that or provide explanation.
I stopped at the figures. I could not continue reviewing the manuscript with the concern about statistics.
Response: The statistics shown on the figures 1-6 were checked and corrected entirely. For example, Figure 5: Water soluble Ca of soil as a function of irrigation water applications during the irrigation periods (1, 2 and 4) and the rainy period (3): a) NaCl water: SAR20; b) NaCl water: SAR40; c) NaHCO3 water: SAR20 and d) NaHCO3 water: SAR40. Within each soil depth (and water type), the columns labelled with same letter are not significantly different at p< 0.05 label.
R4: In my opinion, authors should include the above-mentioned recommendations into the text. It is also necessary to re-write the introduction part to explain the aim of their study.
Response: This part in the Introduction was revised.
Round 2
Reviewer 4 Report
Comments and Suggestions for Authors
For the bar graphs, some comparisons of means and the height of the bars do not match. There are letters not on top of the bars. Bar called b and bar called d, where is bar c? Bar a is taller than bar ab, but shorter than c. There is something wrong with the statistical analysis and interpretations.
Author Response
Dear Reviewers
We would like to thank you for the comment and edition of the manuscript once again.
In the second evaluation, the main issue was related to the statistical problem of the Figures 2-7. Sorry for the confusing. The reviewer #4 might have considered that statistical analysis (for each water quality) was performed based on all the 15 values (15 = 5 period x 3 depth) obtained from all depths. However, the comparison between the periods was made for each soil depth separately (5 values = 5 period x 1 depth), not all depth together.
Figures 2-7 and Supplementary Figures S1 and S2: We have compared the effects of sodic water on a given soil property for each soil depth and displayed it in a single figure (for each water quality). This explanation was provided in the legends of Figures 2-7. We checked the statistical analysis once more and corrected the mistakes, which were mostly related to the shifted letters on the bars. The same-coloured bars represent same soil depth, and the letters on them (a, b, c, ab, etc.) determined by the Duncan tests are concordant among the same-coloured bars. The English language of the article has also been revised considerable.
We made all the corrections on the version that you have edited, and uploaded both versions (the clean and the track changes) of the main manuscript. We hope that the revised manuscript meets with your approval.
Thank you for your consideration.
Sincerely
Prof. Dr. Hasan S. Öztürk
Ankara University
